# No Association between the *SORD* Gene and Amyotrophic Lateral Sclerosis in a Chinese Cohort

**DOI:** 10.3390/jcm11226834

**Published:** 2022-11-18

**Authors:** Mubalake Yilihamu, Ji He, Lu Tang, Yong Chen, Xiaoxuan Liu, Dongsheng Fan

**Affiliations:** 1Department of Neurology, Peking University Third Hospital, Beijing 100191, China; 2Beijing Municipal Key Laboratory of Biomarker and Translational Research in Neurodegenerative Diseases, Beijing 100191, China; 3Key Laboratory for Neuroscience, National Health Commission/Ministry of Education, Peking University, Beijing 100191, China

**Keywords:** amyotrophic lateral sclerosis, Chinese population, *SORD* gene, whole-exome sequencing

## Abstract

Amyotrophic lateral sclerosis (ALS) is a fatal neurodegenerative disorder. Recently a juvenile ALS patient was reported carrying the c.757delG mutation of the sorbitol dehydrogenase (*SORD*) gene, which was also a related mutation of Charcot-Marie-Tooth disease (CMT) and distal hereditary motor neuropathy (dHMN). ALS shares pathogenesis and overlapping genes with CMT and dHMN. We used whole-exome sequencing technology to screen the full-length *SORD* gene in 601 Chinese sporadic ALS patients and 174 controls without a history of neurological diseases. No *SORD* pathogenic variants were identified in the ALS patients. Our current results did not find an association between *SORD* and ALS in Chinese patients, and further studies will be required.

## 1. Introduction

Amyotrophic lateral sclerosis (ALS) is a neurodegenerative disease characterized by rapidly progressing muscle weakness and death due to respiratory failure within two to four years of symptom onset [1]. Approximately 10% of ALS cases are classified as familial ALS (FALS), while 90% of those who do not have affected relatives are considered sporadic ALS cases (SALS) [2]. There are more than 20 genes known to cause ALS, including *SOD1*, *C9orf72*, *TARDBP*, *FUS*, *OPTN*, *DCTN1*, *KIF5A*, *TBK1*, and *SQSTM1* [3]. The mutation in these genes may be found in 50–70% of FALS patients but only in 10% of patients with SALS [4]. Genetic factors are vital to the pathogenesis and development of treatments for ALS. Therefore, we need to learn more about the mutations associated with SALS.

Recently, a homozygous c.757delG mutation of the sorbitol dehydrogenase (*SORD*) gene was identified in a 24-year-old juvenile ALS (JALS) patient [5]. However, whether this JALS patient had familial or sporadic ALS was unclear, as he was adopted by his parents. No additional pathogenic variants of JALS or CMT/dHMN were discovered by next-generation sequencing of two panels of genes associated with (J)ALS and peripheral neuropathies. The *SORD* gene on chromosome 15q21.1 encodes the sorbitol dehydrogenase enzyme, which is involved in the two-step polyol pathway, converting glucose into sorbitol and subsequently into fructose [6]. Defects in the *SORD* gene would lead to a high risk of intracellular sorbitol accumulation. Sorbitol is not only an osmotic stressor but also an oxidative stressor through its sorbitol dehydrogenase reaction [7], playing a role in the pathogenesis of axonal neuropathy and diabetic peripheral neuropathy.

Since biallelic mutations of the *SORD* gene were first reported as the most common causes for hereditary neuropathies [8], *SORD* prevalent mutation c.757delG (p.Ala253Glnfs*27) has been considered one of the most frequent causes of axonal neuropathy, including autosomal recessive axonal Charcot-Marie-Tooth neuropathy (CMT2) and distal hereditary motor neuropathy (dHMN) [9,10,11,12,13,14,15,16,17]. ALS shows high overlap in terms of clinical presentation, targets, and mechanisms of damage with CMT2 and dHMN [18]. Many of the genes involved in CMT2 and dHMN are also associated with ALS, such as the Kinesin family member 5A (*KIF5A*) gene, Dynactin subunit 1 (*DCTN1*) gene, Glycyl-tRNA synthetase (*GARS*) gene, Neurofilament heavy (*NEFH*) gene, and Senataxin (*SETX*) gene [19]. Therefore, it is significant to analyze their overlapping genes. In addition, the *SORD* gene was also proposed as a biomarker candidate gene for another neurodegenerative disease, Alzheimer’s disease (AD), since it is commonly dysregulated between AD blood and brain tissues [20]. Those data imply that *SORD* mutation might be a possible cause of ALS. Therefore, in this study, we sought to investigate the occurrence of *SORD* gene mutations in ALS patients and explore the relationship between *SORD* mutations and ALS clinical phenotypes.

## 2. Materials and Methods

### 2.1. Subjects

The study included 601 Chinese SALS patients registered with the Neurology Department of Peking University Third Hospital from January 2007 to December 2012 and 174 neurologically normal controls. Patients in the case-cohort study were diagnosed with definite, probable, or laboratory-supported probable ALS according to the El Escorial revised criteria [21] by a neurologist specializing in ALS. In this cohort, none of the patients had any symptoms of dementia, and all of them had normal scores on the Edinburgh Cognitive and Behavioral ALS Screen (ECAS) scale [22]. The study was approved by the ethics committee of Peking University Third Hospital. Written informed consent was obtained from all participants. We screened *SOD1*, *TARDBP*, *FUS*, *C9orf72*, *OPTN*, *DCTN1*, *KIF5A*, *TBK1*, and *SQSTM1* genes for all patients before our research.

### 2.2. Mutation Analyses

Genomic DNA samples were extracted from whole blood using standard protocols (Qiagen, Valencia, CA, USA). All 601 SALS patients and 174 controls underwent whole-exome sequencing (WES). We screened the variants by using the Short Genetic Variations Database (dbSNP) (https://www.ncbi.nlm.nih.gov/snp (accessed on 2 October 2022)), the 1000 Genomes Project (1000G) database (http://www.1000genomes.org/ (accessed on 2 October 2022)), gnomAD v2.11 (http://gnomad-sg.org/ (accessed on 2 October 2022)), the China Metabolic Analytics Project (ChinaMAP, www.mBiobank.com (accessed on 2 October 2022)), and the online Chinese Millionome Database (CMDB, https://db.cngb.org/cmdb/ (accessed on 2 October 2022)). The minor allele frequency of the variants was restricted to less than 0.1%. Variants found by WES were further validated by Sanger sequencing and interpreted according to the American College of Medical Genetics and Genomics (ACMG) standards and guidelines [23]. The potential impact of variants was conducted with SIFT (http://sift.jcvi.org (accessed on 2 October 2022)), PolyPhen-2 (http://genetics.bwh.harvard.edu/pph2/ (accessed on 2 October 2022)), MutationTaster software (http://www.mutationtaster.org (accessed on 2 October 2022)), and Combined Annotation Dependent Depletion (CADD, https://cadd.gs.washington.edu/ (accessed on 2 October 2022)), described in our previous studies [10,24].

## 3. Results

This study recruited 601 SALS patients and 174 neurologically normal control individuals. Among them, 225 patients were females, and 376 patients were males; the mean age of onset was 50.61 ± 11.85. Generally, 15.1% of patients were bulbar-onset ALS, whereas 84.9% were spinal-onset ALS; 78.3% of patients were Classic ALS, 8.9% were Bulbar ALS, 7.5% were Flail Arm Syndrome, and 5.3% were Flail leg Syndrome.

The mutation analysis of the *SORD* gene in this study is listed in Table 1. We identified seven nonsynonymous variants in ten SALS patients; among them, five were missense variants (p.Ser82Thr, p.Cys140Ser, p.Ala233Thr, p.Lys243Arg, p.Glu249Lys), one was frameshift variant (p.Ala253Glnfs*27), and the other one was a splice site variant (c.545-6G > C). At the same time, we found another missense variant (p.Val83Leu) and one splice site variant (c.908 + 1G > C) in the two controls. All of the variants were heterozygous. Variants found in SALS patients weren’t found in controls and the other way around. The variant c.908 + 1G > C, identified in controls in this study, was reported as compound heterozygous mutation c.908 + 1G > C/c.404A > G in a Chinese dHMN patient [9]. Different from the homozygous c.757delG (p.Ala253Glnfs*27) mutation previously reported in an ALS patient, we found four SALS patients carrying the heterozygous polymorphism c.757delG (p.Ala253Glnfs*27, rs55901542), but no homozygous or compound heterozygous mutation was found in our subjects. According to the GnomAD v3 database, the heterozygous c.757delG allele count was 623 with an allele frequency of about 0.0043, while the homozygous c.757delG allele count of the *SORD* gene was 1 with an allele frequency of about 0.000007.

Results predicted by in silico tools of nonsynonymous variants of *SORD* were listed in Table 2. The locations and distributions of *SORD* variants in our study are presented in Figure 1B. For the missense variants detected in ALS patients, variant (c.244T > A, p.Ser82Thr) was located in the Zinc binding domain, and variant (c.728A > G, p.Lys243Arg) was located in the tetramer interface domain. The splicing variant c.545-6G > C was located at the intron 5. Human Splicing Finder 3.1 (Genomnis SAS, Marseille, France) predicted this splicing variant as benign. All of the variants were classified as variants of uncertain significance (VUS) according to the standards and guidelines of the ACMG. The clinical characteristics of the SALS patients with *SORD* variants are listed in Table 3.

## 4. Discussion

This is the first study assessing the possible association between the *SORD* gene and ALS. We performed a case-control analysis to determine if *SORD* mutations are associated with ALS in Chinese individuals, and we found no correlation between the *SORD* gene and ALS patients of Chinese descent.

The previously reported French juvenile ALS patient [5] carrying c.757delG homozygous mutation developed lower limb weakness at the age of 21 years old, with a relatively fast course and rapid upper limb involvement. This *SORD* variant is the only variant associated with ALS so far, which is characterized as a homozygous state. Inconsistent with this JALS patient, no c.757delG homozygous mutations nor compound heterozygous mutations were identified in our SALS patients. We only found four SALS patients carrying the heterozygous polymorphism c.757delG (p.Ala253Glnfs*27). Most of these four patients showed an older age at onset and had longer disease duration than this JALS patient. The identification of four heterozygous carriers of this variant, which is relatively common across populations, is not enough to establish a causative role for ALS in these individuals. We hypothesize that the absence of *SORD* mutations in our SALS cohort may be due to the small number of JALS patients (with an onset before the age of 25): 4.8% (29/601). To confirm the possible contribution of *SORD* variants to ALS, more research is required in certain ALS groups, such as the JALS cohort.

The *SORD* is an important enzyme that converts sorbitol to fructose in the polyol pathway. *SORD* deficiency leads to increased levels of tissue and blood sorbitol, cellular osmolarity and oxidative stress, and decreased NADPH levels at the same time [7]. Mutations in *SORD* have been described as the most frequent cause of recessive inherited neuropathies in many studies. Some researchers analyzed the mutation of *SORD* in CMT and dHMN patients. Our team also explored the frequency of *SORD* mutations in CMT and dHMN patients in the previous study. We screened a cohort of 485 unrelated Chinese patients and identified five dHMN patients carrying the *SORD* variant, with the frequency of *SORD* variants being 1% (5/485) in all hereditary neuropathy patients and 6.4% (5/78) in patients with unclarified CMT2 and dHMN [10]. However, the frequency of *SORD* variants in the ALS cohort in our study was 1.7% (10/601), and the frequency in the control cohort was 1.1% (2/174), much lower than that in previous CMT and dHMN studies, which means that the prevalence of *SORD* variants in Chinese ALS patients was lower than that in CMT and dHMN patients. Meanwhile, we did not find an association between ALS and *SORD* in the current study. We speculate that the absence of *SORD* mutations in Chinese SALS patients or other factors could lead to the lack of an association in our cohort.

We have summarized all reported variants of *SORD*-related neuropathy so far in Table 4. We found that the modes of inheritance of *SORD*-related neuropathy were mainly divided into one kind of homozygous mutation and two kinds of compound heterozygous mutations, of which the homozygous mutation was c.757delG (p.Ala253Glnfs*27), and the compound heterozygous mutations were c.757delG (p.Ala253Glnfs*27) with another *SORD* mutations, and c.908 + 1G>C/c.404A > G (p.His135Arg). The phenotypes of patients with *SORD* mutations were that most patients developed their first symptom in adolescence. The schematic graph of the *SORD* protein and all published variants of *SORD*-related neuropathy is shown in Figure 1. There were no significant differences in the regions and domains of variation for different *SORD*-related diseases.

## 5. Conclusions

In conclusion, our study did not find any pathogenic *SORD* variants in the SALS cohort due to the limited sample size. Further studies with more participants or in specific ALS populations, such as the JALS cohort, are needed to validate the potential contribution of *SORD* variants to ALS.

## Figures and Tables

**Figure 1 jcm-11-06834-f001:**
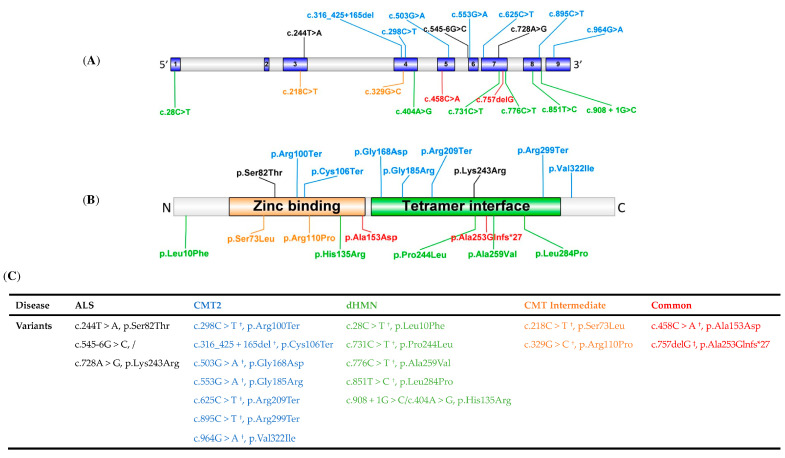
Schematic graph of the SORD protein and published variants. (**A**,**B**) The SORD gene structure; (**C**) overview of the SORD-related neuropathy. Variants identified in our ALS cohort are marked in black; variants identified in CMT2 are marked in blue; variants identified in dHMN are marked in green; variants identified in CMT intermediate are marked in orange; variants identified in both CMT2 and dHMN are marked in red. ^†^ means co-occurrence of SORD c.757delG variant. ^‡^ means homozygous variant and identified in the CMT intermediate as well.

**Table 1 jcm-11-06834-t001:** Descriptions of the nonsynonymous variant of *SORD*.

cDNA	Amino Acid Change	Type	Exon	dbSNP	1000 Genomes	GnomAD East Asian	ChinaMap	SALS	Controls
c.244T > A	p.Ser82Thr	Heterozygous	3	/	/	/	/	1/601	0/174
c.247G > T	p.Val83Leu	Heterozygous	3	/	/	/	/	0/601	1/174
c.418T > A	p.Cys140Ser	Heterozygous	4	rs569483540	/	0.0001087	0.000377786	1/601	0/174
c.545-6G > C	/, splicing	Heterozygous	intron 5–6	/	/	/	/	1/601	0/174
c.697G > A	p.Ala233Thr	Heterozygous	7	rs376874432	/	/	/	1/601	0/174
c.728A > G	p.Lys243Arg	Heterozygous	7	/	/	/	/	1/601	0/174
c.745G > A	p.Glu249Lys	Heterozygous	7	rs776518780	/	/	0.000236116	1/601	0/174
c.757delG	p.Ala253Glnfs*27	Heterozygous	7	rs55901542	/	0.0002528	0.00316396	4/601	0/174
c.908 + 1G > C	/, splicing	Heterozygous	intron 8–9	/	/	/	/	0/601	1/174

Abbreviations: dbSNP, The Single Nucleotide Polymorphism Database; gnomAD, Genome Aggregation Database; SALS, sporadic amyotrophic lateral sclerosis.

**Table 2 jcm-11-06834-t002:** Results predicted by in silico tools of nonsynonymous variants of *SORD*.

cDNA	Variant	Type	SIFT	PolyPhen-2	MutationTaster	CADD	Evidence	ACMG	SALS	Controls
c.244T > A	p.Ser82Thr	Heterozygous	Tolerated	Benign	Polymorphism	4.909	PM_2_, BP4	Uncertain significance	1/601	0/174
c.247G > T	p.Val83Leu	Heterozygous	Deleterious	Probably damaging	Disease-causing	5.800	PM_2_, PP3	Uncertain significance	0/601	1/174
c.418T > A	p.Cys140Ser	Heterozygous	Deleterious	Probably damaging	Disease-causing	25.4	PP3	Uncertain significance	1/601	0/174
c.545-6G > C	/, splicing	Heterozygous	/	/	/	3.039	PM_2_	Uncertain significance	1/601	0/174
c.697G > A	p.Ala233Thr	Heterozygous	Tolerated	Benign	Disease-causing	18.12	PM_2_, BP4	Uncertain significance	1/601	0/174
c.728A > G	p.Lys243Arg	Heterozygous	Tolerated	Benign	Polymorphism	20.2	PM_2_, BP4	Uncertain significance	1/601	0/174
c.745G > A	p.Glu249Lys	Heterozygous	Deleterious	Probably damaging	Disease-causing	26.4	PM_2_, PP3	Uncertain significance	1/601	0/174
c.757delG	p.Ala253Glnfs*27	Heterozygous	/	/	/	/	PVS1	Uncertain significance	4/601	0/174
c.908 + 1G > C	/, splicing	Heterozygous	/	/	/	10.218	PVS1, PM_2_	Uncertain significance	0/601	1/174

Abbreviations: CADD, Combined Annotation Dependent Depletion; ACMG, American College of Medical Genetics and Genomics; SALS, sporadic amyotrophic lateral sclerosis.

**Table 3 jcm-11-06834-t003:** Clinical features of the SALS patients with *SORD* variants.

cDNA	Variant	ID	Sex	Age of Onset (Years)	Site of Onset	Disease Duration (Months)	Clinical Phenotype
c.244T > A	p.Ser82Thr	9113	Female	50	Right hand	46	Classic ALS
c.545-6G > C	/, splicing	8371	Male	58	Right leg	42	Classic ALS
c.728A > G	p.Lys243Arg	7160	Female	56	Right hand	48	Classic ALS
c.757delG	p.Ala253Glnfs*27	7158	Male	53	Left hand	36	Classic ALS
		8366	Male	55	Left leg	15	Classic ALS
		8386	Female	37	Right leg	54	Classic ALS
		8180	Male	56	Left hand	60	Classic ALS

**Table 4 jcm-11-06834-t004:** Summary of clinical features of patients with *SORD* neuropathy.

Inherited Type	cDNA		Amino Acid Change	Count	Phenotype	Sex	Age at Onset (Years)	References
	Allele 1	Allele 2						
Homozygous	c.757del	c.757del	p.Ala253Glnfs*27	14	dHMN	9 male, 5 female	12–40	Cortese et al. [15]
				20	CMT2	13 male, 7 female	10–40	Cortese et al. [15]
				3	CMT intermediate	male	12–25	Cortese et al. [15]
				1	dHMN	male	26	Alluqmani et al. [25]
				2	dHMN	female	4, 14	Wu et al. [26]
				2	CMT2	male, female	17, 16	Yuan et al. [11]
				2	CMT2	male	5, 16	Lin et al. [27]
				2	dHMN	male, female	10, 12	Lin et al. [27]
				1	dHMN	male	10	Laššuthová et al. [14]
				1	CMT intermediate	male	13	Laššuthová et al. [14]
				9	CMT2	5 male, 4 female	0–40	Laššuthová et al. [14]
				3	dHMN	male	9, 10, 15	Dong et al. [9]
				1	dHMN	female	21–30	Frasquet et al. [28]
				2	dHMN	female	17, 6	Liu et al. [10]
Compound heterogeneous	c.757del	c.298C > T	p.Arg100Ter	1	CMT2	male	15	Cortese et al. [15]
		c.329G > C	p.Arg110Pro	1	CMT intermediate	male	13	Cortese et al. [15]
		c.458C > A	p.Ala153Asp	2	CMT2	male, female	10, 20	Cortese et al. [15]
				1	dHMN	female	10–20	Laššuthová et al. [14]
				5	CMT2	3 male, 2 female	0–51	Laššuthová et al. [14]
				1	dHMN	female	2–10	Frasquet et al. [28]
				1	unclear	female	unclear	Frasquet et al. [28]
		c.964G > A	p.Val322Ile	1	CMT2	male	2	Cortese et al. [15]
		c.316_425 + 165del	/	1	CMT2	male	15	Cortese et al. [15]
		c.28C > T	p.Leu10Phe	1	dHMN	male	18	Cortese et al. [15]
		c.895C > T	p.Arg299Ter	1	CMT2	male	15	Cortese et al. [15]
		c.625C > T	p.Arg209Ter	1	CMT2	male	15	Yuan et al. [11]
		c.553G > A	p.Gly185Arg	1	CMT2	male	49	Laššuthová et al. [14]
		c.218C > T	p.Ser73Leu	1	CMT intermediate	male	10–20	Laššuthová et al. [14]
		c.503G > A	p.Gly168Asp	1	CMT2	male	20–25	Laššuthová et al. [14]
		c.776C > T	p.Ala259Val	1	dHMN	male	16	Liu et al. [10]
		c.731C > T	p.Pro244Leu	1	dHMN	female	15	Liu et al. [10]
		c.851T > C	p.Leu284Pro	1	dHMN	male	16	Liu et al. [10]
	c.908 + 1G > C	c.404A > G	p.His135Arg	1	dHMN	male	16	Dong et al. [9]

## Data Availability

The datasets used and/or analyzed during the current study are available from the corresponding author upon reasonable request.

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
