# Peer review of "No Association between the SORD Gene and Amyotrophic Lateral Sclerosis in a Chinese Cohort"

_jcm, 2022, doi:10.3390/jcm11226834_

Round 1

Reviewer 1 Report

In this study the authors reported the frequency of SORD gene variants in an ALS cohort and 174 healthy controls. They could not find any association between ALS patients and variants in SORD gene.

I have some concerns/comments/suggestions:

-          - In the introduction, it would be useful to have more details on the juvenile ALS case already reported in the literature: Was this a familiar or a sporadic case? Were other ALS- associated genes investigated?

-          - Please report the name of genes in italic type.

-          - All 601 ALS patients were defined “without dementia”. How the authors did explore the presence of dementia (only on a clinical impression or through a neuropsychological assessment)? Furthermore, a table summarizing the clinical features of ALS patients included in the study (or otherwise a more detailed explanation in the text) with the following variables (age of onset, ALS phenotype, type of onset) would be appropriate.

-        -  In the Table 2, please add a column indicating if the variant was found in ALS patients or in healthy controls

-         -In the abbreviations of the table 3, “ALSFRS-R, Amyotrophic Lateral Sclerosis Functional Rating Scale–Revised; LMN, lower motor neurons; UMN, upper motor” were reported. However, in the corresponding table there are no information on these variables. Please correct, also adding the clinical phenotype (classic, bulbar, PLMN, PUMN) and the available information not already detailed.

-        - In the discussion the authors stated :” However, the prevalence of SORD mutations in the ALS cohort in our study was much lower than that in previous CMT and dHMN studies, which means that the prevalence of SORD mutations in Chinese ALS patients was lower than that in CMT and dHMN patients”. Please specify the exact frequency of the found variants in your ALS, healthy controls and neuropathy cohorts, in order to add strength to your results.

-          How the authors did explain their findings? Is it possible that their cohort did not include juvenile ALS patients (with an onset before the age of 25?). Please add more clinical information on the ALS cohort and discuss this topic.  

Author Response

(Reviewers can also see the attachment, thank you.)

Dear Editor and reviewers,

Thank you for your letter and the reviewers’ comments concerning our manuscript entitled “An association study between the SORD gene and amyotrophic lateral sclerosis in a Chinese cohort” (Manuscript ID: jcm-2016149). Based on the reviewer’s suggestion, the title of the manuscript is now changed to “No association between the SORD gene and amyotrophic lateral sclerosis in a Chinese cohort”. Those comments are all valuable and very helpful for revising and improving our paper as well as the important guiding significance to our researches. We have taken the reviewers’ comments and suggestions into careful consideration and revised the manuscript accordingly. On the following pages, please find our point-to-point responses to the reviewers’ concerns in the order that they were originally listed, and details of the pages on which the changes have been made.

We had checked all style requirements for Journal of Clinical Medicine one by one carefully. The manuscript has not been submitted nor under consideration for publication by another journal. All authors have read the manuscript and are in agreement that the work is ready for submission and accept the responsibility for manuscript contents. None of the authors have any conflict of interest in the matter.

We believe that the quality of the manuscript has been considerably enhanced as a consequence of the review process. We hope that the revised paper now meets your approval for publication in Journal of Clinical Medicine. Please do not hesitate to contact me if you need any further information.

Sincerely,

Dongsheng Fan

Department of Neurology,

Peking University Third Hospital,

49 North Garden Road, Haidian District, Beijing 100191, People’s Republic of China

Phone: (+86)13701023871

Fax: 086-010-82266250

Email: dsfan2010@aliyun.com

Responds to the reviewers' comments

Reviewer 1

COMMENTS TO AUTHOR(S)

In this study the authors reported the frequency of SORD gene variants in an ALS cohort and 174 healthy controls. They could not find any association between ALS patients and variants in SORD gene.

Response: Thank you for your supportive comments.

Minor Comment

Point 1: In the introduction, it would be useful to have more details on the juvenile ALS case already reported in the literature: Was this a familiar or a sporadic case? Were other ALS- associated genes investigated?

Response 1: Thanks for your thoughtful comments. Whether this juvenile ALS patient is familial or sporadic ALS was unclear, as he was adopted by his parents. No additional pathogenic variants of JALS or CMT/dHMN were discovered by next-generation sequencing of two panels of genes associated with (J)ALS and peripheral neuropathies. We have revised the statement to address your concerns and hope that it is now clearer. Please see page 2, lines 31-34 of the revised manuscript.

Point 2: Please report the name of genes in italic type.

Response 2: Thanks for the reminding. We have revised that point accordingly.

Point 3: All 601 ALS patients were defined “without dementia”. How the authors did explore the presence of dementia (only on a clinical impression or through a neuropsychological assessment)? Furthermore, a table summarizing the clinical features of ALS patients included in the study (or otherwise a more detailed explanation in the text) with the following variables (age of onset, ALS phenotype, type of onset) would be appropriate.

Response 3: Thanks for the thoughtful question. Firstly, patients do not have any symptoms of dementia and have normal scores on the Edinburgh Cognitive and Behavioral ALS Screen (ECAS) scale[1]. Based on the reviewer’s question, we have revised the statement in page 3, lines 55-56 of the revised manuscript. Secondly, the clinical features of ALS patients included in this study have been added in the manuscript in page 4, lines 72-75.

Point 4: In the Table 2, please add a column indicating if the variant was found in ALS patients or in healthy controls.

Response 4: Thank you for your supportive comments. We have added a column indicating whether the variant was found in the ALS cohort or controls both in Table 1 and Table 2.

Point 5: In the abbreviations of the table 3, “ALSFRS-R, Amyotrophic Lateral Sclerosis Functional Rating Scale–Revised; LMN, lower motor neurons; UMN, upper motor” were reported. However, in the corresponding table there are no information on these variables. Please correct, also adding the clinical phenotype (classic, bulbar, PLMN, PUMN) and the available information not already detailed.

Response 5: Thanks for the reminding. We have revised and also added the clinical phenotype in Table 3. All of the SALS patients with SORD variants are Classic ALS.

Point 6: In the discussion the authors stated :” However, the prevalence of SORD mutations in the ALS cohort in our study was much lower than that in previous CMT and dHMN studies, which means that the prevalence of SORD mutations in Chinese ALS patients was lower than that in CMT and dHMN patients”. Please specify the exact frequency of the found variants in your ALS, healthy controls and neuropathy cohorts, in order to add strength to your results.

Response 6: Thank you for your supportive comments. The frequency of SORD variants was 1% (5/485) in all hereditary neuropathy patients and 6.4 % (5/78) in patients with unclarified CMT2 and dHMN [2]. While the frequency of SORD variants in the ALS cohort in our study was 1.7% (10/601), the the frequency in controls cohort was 1.1% (2/174). Based on the reviewer’s suggestion, we have revised the statement in page 7, lines 162-165 of the revised manuscript.

Point 7: How the authors did explain their findings? Is it possible that their cohort did not include juvenile ALS patients (with an onset before the age of 25?). Please add more clinical information on the ALS cohort and discuss this topic.

Response 7: Thanks for the thoughtful question. Our cohort did include juvenile ALS patients. We hypothesize that the absence of SORD mutations in our SALS cohort may be due to the small number of JALS patients (with an onset before the age of 25): 4.8% (29/601). To confirm the possible contribution of SORD variants to ALS, more research is required in certain ALS groups, such as the JALS cohort. We have revised the statement in page 7, lines 153-156 of the revised manuscript.

References

  1. Abrahams, S.; Newton, J.; Niven, E.; Foley, J.; Bak, T.H. Screening for cognition and behaviour changes in ALS. Amyotroph Lateral Scler Frontotemporal Degener 2014, 15, 9-14, doi:10.3109/21678421.2013.805784.
  2. Liu, X.; He, J.; Yilihamu, M.; Duan, X.; Fan, D. Clinical and Genetic Features of Biallelic Mutations in SORD in a Series of Chinese Patients With Charcot-Marie-Tooth and Distal Hereditary Motor Neuropathy. Front Neurol 2021, 12, 733926, doi:10.3389/fneur.2021.733926.

Reviewer 2 Report

Mubalake Yilihamu and colleagues  analyzed  SORD gene, recently associated to ALS, in 601 Chinese sporadic ALS patients and 174 controls. No pathogenic variants were found in patients, suggesting no association between SORD gene and ALS. The study reports negative results and this should be clear in the title. I would suggest modifying it in "No association between the SORD gene and amyotrophic lateral sclerosis in a Chinese cohort". Furthermore, a few modifications need to be done. 

- Gene symbols have to be written in italics, according to international guidelines.

-       Please modify the term “mutation” into “genetic variant”, as established by ACMG nomenclature.

If segregation analysis of identified variant is possible, it would be useful. 

Author Response

(Reviewers can also see the attachment, thank you.)

Dear Editor and reviewers,

Thank you for your letter and the reviewers’ comments concerning our manuscript entitled “An association study between the SORD gene and amyotrophic lateral sclerosis in a Chinese cohort” (Manuscript ID: jcm-2016149). Based on the reviewer’s suggestion, the title of the manuscript is now changed to “No association between the SORD gene and amyotrophic lateral sclerosis in a Chinese cohort”. Those comments are all valuable and very helpful for revising and improving our paper as well as the important guiding significance to our researches. We have taken the reviewers’ comments and suggestions into careful consideration and revised the manuscript accordingly. On the following pages, please find our point-to-point responses to the reviewers’ concerns in the order that they were originally listed, and details of the pages on which the changes have been made.

We had checked all style requirements for Journal of Clinical Medicine one by one carefully. The manuscript has not been submitted nor under consideration for publication by another journal. All authors have read the manuscript and are in agreement that the work is ready for submission and accept the responsibility for manuscript contents. None of the authors have any conflict of interest in the matter.

We believe that the quality of the manuscript has been considerably enhanced as a consequence of the review process. We hope that the revised paper now meets your approval for publication in Journal of Clinical Medicine. Please do not hesitate to contact me if you need any further information.

Sincerely,

Dongsheng Fan

Department of Neurology,

Peking University Third Hospital,

49 North Garden Road, Haidian District, Beijing 100191, People’s Republic of China

Phone: (+86)13701023871

Fax: 086-010-82266250

Email: dsfan2010@aliyun.com

Responds to the reviewers' comments

Reviewer 2

COMMENTS TO AUTHOR(S)

Mubalake Yilihamu and colleagues analyzed SORD gene, recently associated to ALS, in 601 Chinese sporadic ALS patients and 174 controls. No pathogenic variants were found in patients, suggesting no association between SORD gene and ALS. The study reports negative results and this should be clear in the title. I would suggest modifying it in "No association between the SORD gene and amyotrophic lateral sclerosis in a Chinese cohort". Furthermore, a few modifications need to be done.

Response: Thank you for your supportive suggestion. Based on the reviewer’s suggestion, we have revised the title of manuscript as "No association between the SORD gene and amyotrophic lateral sclerosis in a Chinese cohort".

Minor Comment

Point 1: Gene symbols have to be written in italics, according to international guidelines.

Response 1: Thanks for the reminding. We have revised that point accordingly.

Point 2: Please modify the term “mutation” into “genetic variant”, as established by ACMG nomenclature.

Response 2: Thanks for your thoughtful comments. We have modified that point accordingly in the full text.

Point 3: If segregation analysis of identified variant is possible, it would be useful. 

Response 3: Thank you very much for your valuable suggestions. Due to the unwillingness of families in this study, we were not able to perform the separation analysis, in our future follow-up studies we will involve.

Round 2

Reviewer 1 Report

The authors addressed all my comments/suggestions, and the article is now suitable for publication according to my opinion.

I would just point out some errors in the current version:

-          Page 4: Bublar instead of Bulbar phenotype

-          Page 7, line 127: the article “the” has been typed twice.

Author Response

(Reviewers can also see the attachment, thank you.)

Dear Editor and reviewers,

Thank you for your letter and the reviewers’ comments concerning our first manuscript entitled “An association study between the SORD gene and amyotrophic lateral sclerosis in a Chinese cohort” (Manuscript ID: jcm-2016149). Based on the reviewer’s suggestion, the title of the manuscript is now changed to “No association between the SORD gene and amyotrophic lateral sclerosis in a Chinese cohort”. Those comments are all valuable and very helpful for revising and improving our paper as well as the important guiding significance to our researches. We have taken the reviewers’ comments and suggestions into careful consideration and revised the manuscript accordingly. On the following pages, please find our point-to-point responses to the reviewers’ concerns in the order that they were originally listed, and details of the pages on which the changes have been made.

We had checked all style requirements for Journal of Clinical Medicine one by one carefully. The manuscript has not been submitted nor under consideration for publication by another journal. All authors have read the manuscript and are in agreement that the work is ready for submission and accept the responsibility for manuscript contents. None of the authors have any conflict of interest in the matter.

We believe that the quality of the manuscript has been considerably enhanced as a consequence of the review process. We hope that the revised paper now meets your approval for publication in Journal of Clinical Medicine. Please do not hesitate to contact me if you need any further information.

Sincerely,

Dongsheng Fan

Department of Neurology,

Peking University Third Hospital,

49 North Garden Road, Haidian District, Beijing 100191, People’s Republic of China

Phone: (+86)13701023871

Fax: 086-010-82266250

Email: dsfan2010@aliyun.com

Responds to the reviewers' comments

Reviewer 1

COMMENTS TO AUTHOR(S)

The authors addressed all my comments/suggestions, and the article is now suitable for publication according to my opinion.

Response: Thank you so much for your supportive comments.

Minor Comment

Point 1: I would just point out some errors in the current version: Page 4: Bublar instead of Bulbar phenotype.

Response 1: Thanks for your thoughtful comments. We have corrected this typo. Please see page 4, lines 74 of the revised manuscript.

Point 2: Page 7, line 127: the article “the” has been typed twice.

Response 2: Thanks for the reminding. We have revised that point accordingly.

We tried our best to improve the manuscript and made some changes in the manuscript. These changes will not influence the content and framework of the paper. Moreover, here we did not list the changes but marked in red lines in the revised manuscript. We would like to thank the referee again for taking the time to review our manuscript.
